# Langevin Quasi-Monte Carlo

**Sifan Liu**
Department of Statistics
Stanford University
Stanford, CA 94305
`sfliu@stanford.edu`

## Abstract

Langevin Monte Carlo (LMC) and its stochastic gradient versions are powerful algorithms for sampling from complex high-dimensional distributions. To sample from a distribution with density $\pi(\theta) \propto \exp(-U(\theta))$, LMC iteratively generates the next sample by taking a step in the gradient direction $\nabla U$ with added Gaussian perturbations. Expectations w.r.t. the target distribution $\pi$ are estimated by averaging over LMC samples. In ordinary Monte Carlo, it is well known that the estimation error can be substantially reduced by replacing independent random samples by quasi-random samples like low-discrepancy sequences. In this work, we show that the estimation error of LMC can also be reduced by using quasi-random samples. Specifically, we propose to use completely uniformly distributed (CUD) sequences with certain low-discrepancy property to generate the Gaussian perturbations. Under smoothness and convexity conditions, we prove that LMC with a low-discrepancy CUD sequence achieves smaller error than standard LMC. The theoretical analysis is supported by compelling numerical experiments, which demonstrate the effectiveness of our approach.

## 1 Introduction

Sampling from probability distributions is a crucial task in both statistics and machine learning. However, when the target distribution does not permit exact sampling, researchers often rely on Markov chain Monte Carlo (MCMC) methods. These techniques simulate a Markov chain that converges to the target distribution as its stationary distribution. Recently, MCMC samplers based on discretizing the continuous-time Langevin diffusion have become popular, due to its ease of implementation and ability to handle stochastic gradients (Welling and Teh, 2011).

The primary focus of this work is on the quality of samples generated by Langevin Monte Carlo (LMC) algorithms in terms of estimating the expectation $\mathbb{E}_{\theta \sim \pi}[f(\theta)]$ for some integrand $f$ by sample averages. In the context of Bayesian inference, the target distribution $\pi$ is typically the posterior distribution, and computing the posterior expectation, posterior variance, or confidence intervals are of great interest. In the context of post-selection inference, the target distribution $\pi$ is the probability distribution conditioned on the selection event, and computing the selection-adjusted p-value is the main task. LMC has been widely used in this problem as well (Markovic and Taylor, 2016; Shi et al., 2022). In all these situations, the accuracy of the sample average estimator is critical and affects the downstream data analysis.

In traditional Monte Carlo sampling, it is well known that using quasi-Monte Carlo (QMC) samples, instead of independent and identically distributed (i.i.d.) random samples, can lead to significant error reduction. So it is natural to ask whether we can apply QMC techniques to improve Langevin Monte Carlo sampling as well. In this work, we introduce the Langevin quasi-Monte Carlo (LQMC) algorithm, which replaces the i.i.d. random inputs in the LMC algorithm with quasi-random numbers.

37th Conference on Neural Information Processing Systems (NeurIPS 2023).

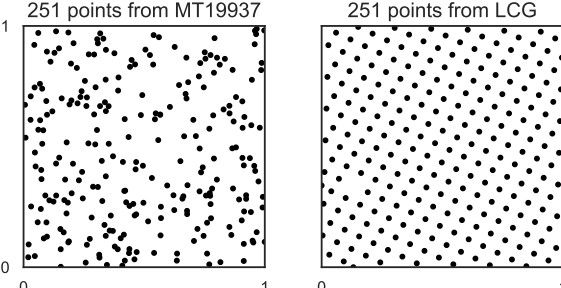

Figure 1: Scatter plots of 251 points generated from Mersenne Twister 19937 (left) and 251 points generated from a linear congruential generator (LCG) of period 251. Points from an entire period of a pseudo-random number generator (right) fill the unit square more evenly than the same number of points from a PRNG with a larger period (left).

These quasi-random numbers are carefully designed to sample from the target distribution more evenly and more balanced, leading to improved estimation accuracy.

Not all quasi-Monte Carlo point sets are suitable for simulating Markov chains. Suppose the Markov chain is driven by a sequence of uniform random vectors in the unit cube. A sufficient condition for the sequence is known as *completely uniformly distributed* (CUD). In our implementation of the driving sequence, we use an entire period of a pseudo-random number generator (PRNG). While modern computer simulations often use PRNGs with a large period, such as Mersenne Twister with a period of $2^{19937} - 1$, our approach runs through the entire period of a PRNG with a relatively small period in the LMC algorithm. The advantage of using an entire period of a PRNG is that the points are more evenly distributed, which is more desirable for numerical integration. We illustrate the balancing property of an entire PRNG in Figure 1.

The main contributions of this paper are threefold. First, we propose a novel technique of using quasi-random numbers in Langevin-type algorithms, which can be applied to a wide range of such algorithms by substituting i.i.d. random numbers with a sequence of quasi-random numbers. The quasi-random numbers are constructed similarly as usual PRNGs, therefore no extra computational complexity is required. Second, we evaluate the performance of the proposed LQMC algorithm in a variety of numerical experiments, demonstrating that it can significantly reduce the mean squared error (MSE) of traditional LMC by a factor ranging from 2 to 500, depending on the problem. Finally, we provide theoretical analysis showing that LQMC can reduce the Monte Carlo part of the error from $O(n^{-1/2})$ to $O(n^{-1+\delta})$ for any $\delta > 0$ in situations where the Markov chain is strongly contracting and the integrand function $f$ is sufficiently regular. This error reduction is consistent with the usual improvement achieved by using quasi-Monte Carlo in place of plain Monte Carlo.

The rest of the paper is organized as follows. In Section 2, we provide some background on LMC and QMC, followed by a review of related work. Section 3 describes the LQMC algorithm and its implementation details. In Section 4, we present theoretical guarantees for the proposed method. Finally, in Section 5, we provide empirical results to evaluate the performance of LQMC and compare it with the standard LMC algorithm.

## 2 Backgrounds

This section provides some background on Langevin Monte Carlo and quasi-Monte Carlo.

### 2.1 Langevin Monte Carlo

Suppose we want to sample from the target distribution $\pi(\theta) \propto \exp(-U(\theta))$ where $\theta \in \mathbb{R}^d$ and $U$ is known as the potential function. LMC algorithms are based on Euler-Maruyama discretization of the Langevin diffusion $\theta(t)$, which satisfies the stochastic differential equation

$$\mathrm{d}\theta(t) = -\nabla U(\theta(t))\mathrm{d}t + \sqrt{2}\mathrm{d}W_t, \tag{1}$$

where $\{W_t\}_{t\geq 0}$ is a $d$-dimensional standard Brownian motion. Under mild technical conditions, the Langevin diffusion $\theta(t)$ has $\pi$ as its unique invariant distribution (Roberts and Tweedie, 1996). With a discretization step size $h$, LMC updates the sample $\theta_k$ by

$$\theta_{k+1} \leftarrow \theta_k - h\nabla U(\theta_k) + \sqrt{2h}\xi_{k+1} \tag{2}$$

where $\xi_k \overset{iid}{\sim} \mathcal{N}(0, I_d)$.

In many applications, we are interested in computing the expectation $\mu := \mathbb{E}_{\theta\sim\pi}[f(\theta)]$ over $\pi$ for some $\pi$-integrable function $f$. The LMC estimator of $\mu$ is the sample average

$$\hat{\mu}_n = \frac{1}{n}\sum_{k=1}^{n} f(\theta_k),$$

where $n$ is the number of iterations.

Teh et al. (2016) provide an asymptotic bias-variance decomposition of the MSE of the weighted average $\frac{\sum_{k=1}^{n} h_k f(\theta_k)}{\sum_{k=1}^{n} h_k}$ and show that the optimal step size scales as $h_k \asymp k^{-1/3}$, leading to an MSE of order $O(n^{-2/3})$. Here $h_k$ is the step size used at the $k$-th iteration. Vollmer et al. (2016) generalize this result to the non-asymptotic setting with a constant step size $h$. They show that the MSE is of order $O(h^2 + \frac{1}{nh})$, where $h^2$ corresponds to the squared bias and $\frac{1}{nh}$ corresponds to the variance.

## 2.2 Quasi-Monte Carlo

QMC is an alternative to Monte Carlo for numerical integration and is well-known for having much higher accuracy than Monte Carlo. QMC is primarily designed to numerically evaluate the integral $\mu = \int_{[0,1]^d} f(\mathbf{u})\mathrm{d}\mathbf{u}$. It estimates $\mu$ by taking points $\mathbf{u}_i \in [0,1]^d$ and let the estimator be

$$\hat{\mu} = \frac{1}{n}\sum_{i=1}^{n} f(\mathbf{u}_i).$$

Unlike Monte Carlo which takes $\mathbf{u}_i$ to be identically independently distributed (i.i.d.), QMC constructs the point set $\{\mathbf{u}_i\}_{i=1}^{n}$ that aims to minimize the star discrepancy

$$D_n^* = D_n^*(\mathbf{u}_1,\ldots,\mathbf{u}_n) = \sup_{\mathbf{a}\in[0,1]^d}\left|\frac{1}{n}\sum_{i=1}^{n}1\{\mathbf{u}_i \in [\mathbf{0},\mathbf{a})\} - \prod_{j=1}^{d} a_j\right|. \tag{3}$$

The star discrepancy measures the uniformity of the point sets by comparing the fraction of points inside $[\mathbf{0},\mathbf{a})$ and the volume $\prod_{j=1}^{d} a_j$, taking supreme over all the rectangles inside $[0,1]^d$ anchored at $\mathbf{0}$. QMC can generate points with $D_n^* = O(n^{-1}(\log n)^{d-1})$, thus QMC is also known as low-discrepancy sequence. Commonly used QMC points include Sobol' sequence (Sobol', 1967), Niederreiter's sequence (Niederreiter, 1987), Halton's sequence, and lattice rules. For a comprehensive survey, we refer to the monograph Dick and Pillichshammer (2010). If the integrand $f$ has bounded variation in the sense of Hardy and Krause $\|f\|_{\mathrm{HK}}$, then the Koksma-Hlawka inequality (see e.g. Dick and Pillichshammer (2010)) bounds the integration error by

$$|\hat{\mu} - \mu| \leq D_n^* \cdot \|f\|_{\mathrm{HK}} \leq O(n^{-1}(\log n)^{d-1}). \tag{4}$$

While the Koksma-Hlawka inequality shows that QMC is asymptotically better than usual Monte Carlo, it doesn't provide a practical way to estimate the error. Moreover, integrands might have infinite Hardy-Krause variation.

One can apply randomization techniques to QMC to address both problems. Common randomization techniques include random shifts (Cranley and Patterson, 1976) and scrambling (Owen, 1995). For RQMC samples $\mathbf{u}_1,\ldots,\mathbf{u}_n$, each $\mathbf{u}_i \sim \mathrm{Unif}([0,1]^d)$ individually but they have the low-discrepancy property collectively with probability 1. One can estimate the error by multiple independent random replicates. For sufficiently smooth $f$, the scrambled Sobol' sequence has variance $O(n^{-3}(\log n)^{d-1})$ (Owen, 1997a,b).

## 2.3 Related work

The first attempt to apply quasi-random numbers to simulate stochastic differential equations was made by Hofmann and Mathé (1997). They showed that if a numerical scheme is weakly convergent with i.i.d. samples, then using completely uniformly distributed (CUD) sequences also leads to consistent estimation. They also demonstrated that certain low-discrepancy sequences are not suitable for simulating SDEs. There have also been some efforts to apply QMC to MCMC. Owen and Tribble (2005) proposed to apply CUD sequences to a Metropolis algorithm and showed that the method is consistent in problems with finite state spaces. Chen et al. (2011) generalized the consistency result to continuous state spaces under the assumption that the Markov chain is a contraction. More recently, Dick et al. (2016); Dick and Rudolf (2014) proved that there exists constructions of the driving sequence $\{\mathbf{u}_k\}_{k \geq 1}$ such that the discrepancy between the empirical distribution of MCMC samples and the target distribution is bounded by $O(n^{-1/2}(\log n)^{1/2})$, the same rate achieved by random inputs. Another line of applying QMC to Markov chains is known as array-RQMC proposed by L'Ecuyer et al. (2008). Array-RQMC runs in parallel multiple Markov chains, and each iteration involves a complicated reordering of the states so that the low-discrepancy among the chains is maintained. Empirically, it achieves significantly smaller estimation error than usual MCMC, but theoretical guarantees remain a challenging open problem.

There has been a growing interest in using QMC techniques in various machine learning tasks, such as variational inference (Buchholz et al., 2018; Liu and Owen, 2021), policy learning and evaluation (Arnold et al., 2022), reinforcement learning with evolution strategies (Choromanski et al., 2019; Rowland et al., 2018), compression of large datasets (Dick and Feischl, 2021), example selection in stochastic gradient descent (SGD) (Lu et al., 2021), and deep learning for solving partial differential equations (Longo et al., 2021).

Numerous efforts have been devoted to improving LMC and stochastic gradient Langevin dynamics (SGLD). To overcome the instability of Euler-Maruyama discretization, various numerical schemes have been proposed, including higher-order integrators (Chen et al., 2015), underdamped LMC (Cheng et al., 2018), and stochastic Runge-Kutta diffusion (Li et al., 2019). For SGLD, variance reduction techniques such as SAGA and SVGR (Dubey et al., 2016) and control variates (Baker et al., 2019) have been proposed. LMC also provides a useful perspective for optimization, as demonstrated by the analyses in Chen et al. (2016); Dalalyan (2017); Raginsky et al. (2017); Xu et al. (2018); Erdogdu et al. (2018). Our contribution is orthogonal to all the aforementioned work, as our algorithm only modifies the random numbers used in the algorithm. Therefore, our method can be combined with other algorithms without interference.

## 3 QMC for LMC

In the LMC algorithm, we can think of the Markov chain as being driven by a sequence of uniform variables $\mathbf{u}_k$ in the unit cube $[0, 1]^d$. For instance, the Gaussian perturbation can be represented as $\xi_k = \Phi^{-1}(\mathbf{u}_k)$, where $\Phi^{-1}$ denotes the inverse Gaussian CDF applied element-wise to $\mathbf{u}_k$. If a stochastic gradient is employed, the randomness associated with the stochastic gradient can also be expressed as uniform variables. Therefore, we can write the transition of the Markov chain as $\theta_{k+1} = \psi(\theta_k, \mathbf{u}_{k+1})$. In typical computer experiments, $\mathbf{u}_k$ are not really i.i.d. but are deterministic pseudo-random numbers. In this section, we will describe an alternative method of generating the pseudo-random numbers $\mathbf{u}_k$, which are carefully constructed and can lead to more accurate sample averages.

The idea here is to use point sets that are more evenly distributed such as QMC points, which can lead to significant improvement in the usual Monte Carlo estimation. However, caution is required when using QMC points to simulate an SDE like (1). This is because the correlation between successive QMC samples may introduce undesired behavior in the Markov chain, as demonstrated in (Tribble, 2007, Section 3.2). To avoid the dependence among successive values, we require that the blocks of points $(v_i, v_{i+1}, \ldots, v_{i+d-1})$ for any lag $d$ are uniformly distributed. This notion of uniformity is formally known as completely uniformly distributed (CUD, Korobov (1948)), which we define next.

We say an infinite sequence $\{\mathbf{u}_i\}_{i=1}^{\infty} \subseteq [0, 1]^d$ is uniformly distributed on $[0, 1]^d$ if the star discrepancy $D^*(\{\mathbf{u}_i\}_{i=1}^n)$ goes to 0 as $n \to \infty$, where the star discrepancy is defined in Equation 3.

**Definition 3.1** (Completely uniformly distributed sequence (CUD)). *An infinite sequence $\{v_i\}_{i=0}^{\infty} \subset [0,1]$ is called completely uniformly distributed, if for all positive integer $d$, the sequence $\{(v_k, \ldots, v_{k+d-1})\}_{k=0}^{\infty} \subseteq \mathbb{R}^d$ is uniformly distributed on $[0,1]^d$. A triangular array $\mathbf{v}_n = (v_{n,1}, \ldots, v_{n,N_n})$ is called array-CUD, if for all positive integer $d$, $D^*((v_{n,1}, \ldots, v_{n,d}), (v_{n,2}, \ldots, v_{n,d+1}), \ldots, (v_{n,N_n-d+1}, \ldots, v_{n,N_n})) \to 0$ as $n \to \infty$, $N_n \to \infty$.*

In other words, the subsequent $d$-tuples in a CUD sequence are uniformly distributed in the $d$-dimensional unit cube for any positive dimension $d$. Now we are ready to present the main algorithm.

## 3.1 LQMC algorithm

Let $\{v_i\}_{i=0}^{\infty}$ be a CUD sequence. Let $\mathbf{u}_k = (v_{kd}, \ldots, v_{(k+1)d-1}) \in \mathbb{R}^d$ be the $k$-th non-overlapping $d$-tuple from the sequence ($k \geq 0$). A CUD sequence is often constructed deterministically. They can further be randomized using the Cranley-Patterson (i.e. random shift) rotation (Cranley and Patterson, 1976)

$$\mathbf{u}_k \leftarrow \mathbf{u}_k + \Delta \mod 1,$$

where $\Delta \sim \text{Unif}([0,1]^d)$. The Cranley-Patterson rotation randomly shifts each dimension of $\mathbf{u}_k$ by a uniform random number separately. Then each $\mathbf{u}_k$ is uniformly distributed on $[0,1]^d$. If we apply the inverse Gaussian CDF to each coordinate of $\mathbf{u}_k$, then $\Phi^{-1}(\mathbf{u}_k) \sim \mathcal{N}(0, I_d)$. In the Langevin-type algorithms, we will let $\xi_k = \Phi^{-1}(\mathbf{u}_k)$ and use $\xi_k$ as the Gaussian perturbation in the $k$-th iteration. Specifically, each iteration takes the form

$$\theta_{k+1} = \theta_k - h\nabla U(\theta_k) + \sqrt{2h} \cdot \Phi^{-1}(\mathbf{u}_{k+1}), \quad k \geq 0.$$

Thus the transition map is $\psi(\theta, \mathbf{u}) = \theta - h\nabla U(\theta) + \sqrt{2h}\Phi^{-1}(\mathbf{u})$. In practice, we can only run finite many iterations. In the following, we will describe how to construct a finite CUD sequence and feed it into the LMC algorithm.

## 3.2 Construction of CUD sequences

A finite CUD (array-CUD) sequence is often implemented by using an entire period of a pseudo random number generator with a small period (Tribble, 2007). There exist other constructions of CUD sequences. For further details, interested readers can refer to Levin (1999). We propose to use the linear-feedback shift register (LFSR) provided in Chen (2011), because it has demonstrated good performance and the computational effort required is comparable to other commonly used PRNGs.

The binary Galois LFSR (Tausworthe generator, Tausworthe (1965)) of order $m$ updates the states $b_i \in \{0, 1\}$ recursively by

$$b_i = \sum_{j=0}^{m-1} a_j b_{i-m+j} \mod 2, \quad i \geq m$$

with initial states $b_0, b_1, \ldots, b_{m-1}$ pre-specified. The $m$-tuple $(b_i, b_{i+1}, \ldots, b_{i+m-1}) \in \text{GF}(2)^m$ can only take $2^m$ different values. If there is an $m$-tuple that is all zero, then all $b_i$'s in this sequence must be zero. So the period of the sequence $\{b_i\}_{i \geq 0}$ is at most $n = 2^m - 1$. Moreover, the period is exactly equal to $2^m - 1$ if and only if the characteristic polynomial

$$x^m + a_{m-1}x^{m-1} + \ldots + a_1 x + a_0$$

is a primitive polynomial over GP(2) (Niederreiter, 1992, Lemma 9.1). Given the states $\{b_i\}_{i \geq 0}$ and an offset $s > 0$ such that $\gcd(s, 2^m - 1) = 1$, $v_i$ is computed with

$$v_i = \sum_{j=0}^{m-1} b_{si+j} 2^{-j-1}, \quad i = 0, 1, \ldots, 2^m - 2.$$

That is, for each $i$, we take the $m$-tuple $(b_{si+j})_{0 \leq j < m}$ and interpret it as the binary expansion of $v_i$. For the next step, we jump $s$ bits ahead in the sequence $\{b_i\}_{i \geq 0}$ and use the $m$-tuple starting from $b_{s(i+1)}$. Chen (2011) provided a table of the LFSR generators for $10 \leq m \leq 32$. They searched the offsets so that the LFSR has good equi-distributed properties. Our experiments use the LFSR generators listed there.

Given the sequence $\{v_i\}_{i=0}^{n-1}$ of length $n$, we repeat it $d$ times and arrange $v_i$'s in the following $n \times d$ matrix

$$\begin{pmatrix} v_0 & v_1 & \cdots & v_{d-1} \\ v_d & v_{d+1} & \cdots & v_{2d-1} \\ \vdots & \vdots & \ddots & \vdots \\ v_{(n-1)d} & v_{(n-1)d+1} & \cdots & v_{nd-1} \end{pmatrix}. \tag{5}$$

We run the LMC algorithm $n = 2^m - 1$ iterations. The $k$-th uniform vector $\mathbf{u}_k$ is the $k$-th row of the above matrix. The procedure is summarized in Algorithm 1.

---

**Algorithm 1** Langevin quasi-Monte Carlo (LQMC)

---

**Input:** Number of iterations $n = 2^m - 1$ such that $\gcd(2^m - 1, d) = 1$, step size $h$, initial value $\theta_0$
  Generate an LFSR sequence $\{v_i\}_{i \geq 0}$ of period $2^m - 1$.
  Let $\mathbf{u}_k = (v_{(k-1)d}, \ldots, v_{kd-1}) \in [0, 1]^d$, for $1 \leq k \leq n$.
  Apply Cranley-Patterson rotation (random shift) to $\mathbf{u}_k$'s.
  **for** $k \leftarrow 1, \ldots, n$ **do**
    $\theta_k \leftarrow \theta_{k-1} - h\nabla U(\theta_{k-1}) + \sqrt{2h}\Phi^{-1}(\mathbf{u}_k)$
  **end for**
**Output:** $\theta_1, \ldots, \theta_n$

---

If $\gcd(n, d) = 1$, then each column of the matrix (5) contains no repeated values. This means that among the $n = 2^m - 1$ iterations of the LQMC algorithm, each dimension uses one value in each sub-interval $\left(\frac{k}{2^m}, \frac{k+1}{2^m}\right]$ at most once ($0 \leq k \leq 2^m - 1$). This perfect one-dimensional stratification is one of the reasons why CUD may achieve smaller estimation error than pseudo-random numbers. If $\gcd(n, d) > 1$, then we take $d'$ to be the smallest integer greater than $d$ and co-prime with $n$. We then create the matrix in (5) similarly but with $d'$ columns. In the LQMC algorithm, we take $\mathbf{u}_k$ to be the $k$-th row of the matrix but only use the first $d$ coordinates.

Algorithm 1 may seem to be restricted by having a fixed number of iterations, $n = 2^m - 1$. However, in practice, the LQMC algorithm can be started with an initial value of $m$. If the chain does not converge after $2^m - 1$ iterations, one can continue the chain with another freshly generated LFSR, possibly with a larger period. This allows for flexibility in adjusting the number of iterations based on the convergence of the chain. Additionally, if a burn-in period is required, one can first run the algorithm with an LFSR of a small period to serve as the burn-in stage and then continue with a larger LFSR. Furthermore, running multiple chains with independent random shifts is embarrassingly parallel. We present the algorithm in the form of the basic LMC algorithm with accurate gradient and constant learning rate. However, as we noted previously, other Langevin-type algorithms can also utilize the CUD sequence directly by substituting the pseudo-random numbers with the LFSR sequence.

## 4    Theoretical guarantee

Here we study the estimation error $|n^{-1} \sum_{k=1}^{n} f(\theta_k) - \pi(f)|$ of LQMC for some test function $f$ that is 1-Lipschitz and bounded. As the first attempt to prove the convergence rate of using QMC in LMC, we impose the relatively strong conditions of smoothness and convexity.

**Assumption 1.** *The potential function $U$ is $L$-smooth*

$$\|\nabla U(\theta) - \nabla U(\theta')\|_2 \leq L\|\theta - \theta'\|_2, \quad \forall \theta, \theta',$$

*and $M$-strongly convex*

$$U(\theta') \geq U(\theta) + \nabla U(\theta)^\mathsf{T}(\theta' - \theta) + \frac{M}{2}\|\theta' - \theta\|_2^2, \quad \forall \theta, \theta'.$$

We will also assume a constant step size $h$. While LMC with vanishing step sizes converges weakly to the target distribution, in practice a constant step size is often used (Vollmer et al., 2016; Brosse et al., 2018). With a constant step size, we can derive a non-asymptotic error bound for LQMC.

Assumption 1 implies that if the step size $h \leq \frac{2}{L+M}$, then the transition map $\psi$ is a strong contraction with parameter $\rho = 1 - hM$, i.e.

$$\|\psi(\theta, \mathbf{u}) - \psi(\theta', \mathbf{u})\|_2 = \|\theta - \theta' - h(\nabla U(\theta) - \nabla U(\theta'))\|_2 \leq \rho \|\theta - \theta'\|_2. \tag{6}$$

See e.g. Lemma 2 of Dalalyan and Karagulyan (2019). The strong contraction implies that if we start two chains from $\theta$ and $\theta'$, and use the same random numbers at every step, then the two chains will merge exponentially fast. In other words, the state $\theta_k$ largely depends on the most recent iterations and quickly forgets about the past history. Formally, let $\mathbf{w}_k^{(\ell)} = (\mathbf{u}_k, \ldots, \mathbf{u}_{k-\ell+1})$ denote the random numbers used in the most recent $\ell$ steps. Define the $\ell$-step transition as

$$\theta_k = \psi_\ell(\theta_{k-\ell}, \mathbf{w}_k^{(\ell)})$$

and let $\bar{f}_\ell(\mathbf{w}_k^{(\ell)})$ denote the value of $f(\theta_k)$ marginalized over $\theta_{k-\ell} \sim \pi$, i.e.

$$\bar{f}_\ell(\mathbf{w}_k^{(\ell)}) = \int f \circ \psi_\ell(x, \mathbf{w}_k^{(\ell)}) \pi(\mathrm{d}x).$$

Thus $\bar{f}_\ell(\mathbf{w}_k^{(\ell)})$ only depends on the most recent $\ell$ iterations. Due to the strong contraction, $|\bar{f}_\ell(\mathbf{w}_k^{(\ell)}) - f(\theta_k)|$ decays exponentially fast with $\ell$. So for large $\ell$, the estimation error of $n^{-1} \sum_{k=1}^n f(\theta_k)$ is close to the error of $\frac{1}{n-\ell} \sum_{k=\ell+1}^n \bar{f}_\ell(\mathbf{w}_k^{(\ell)})$. The latter can be viewed as a $d\ell$-dimensional numerical integration scheme based on the point set $\{\mathbf{w}_k^{(\ell)}\}_{k=\ell+1}^n$. By leveraging the discrepancy bound of the LFSR sequence and assuming that $\bar{f}_\ell$ has bounded variation in the sense of Hardy and Krause, we can derive an error bound using the Koksma-Hlawka inequality (4). Now we state the main error bound and leave the detailed proof in the Appendix A.

**Theorem 4.1** (Error bound of LQMC). *Let Assumption 1 hold. Define the step size $h \leq \frac{2}{L+M}$, $\rho = 1 - hM$, $\ell = \lceil (1/2) \log_\rho h \rceil$. Let $\theta_1, \ldots, \theta_n$ be the output of Algorithm 1 which runs $n$ iterations with step size $h \leq \frac{2}{L+M}$. Assume the LFSR sequence $\{v_i\}_{i \geq 0}$ in use has period $n = 2^m - 1$, offset $s$, and $gcd(m, n) = \gcd(d\ell, n) = 1$. If $\bar{f}_\ell$ has bounded variation in the sense of Hardy and Krause, then as $n \to \infty$ we have*

$$\left| \frac{1}{n} \sum_{k=1}^n f(\theta_k) - \pi(f) \right| \leq C_1 n^{-1+\delta} + C_2 h^{1/2}, \quad \forall \delta > 0.$$

*Here $\delta$ hides poly-logarithmic factors $(\log n)^d$, $C_1$ depends on $d, \ell$ and $\|\bar{f}_\ell\|_{HK}$, and $C_2 = \frac{3\sqrt{2}}{2} \frac{L}{M} d + \max_{0 \leq k \leq n} \|\theta_k\| + \mathbb{E}_\pi [\|\theta\|]$.*

The upper bound consists of two terms. The first term represents the numerical integration error, which arises from the discrepancy of the point set used in the integration scheme. By utilizing low-discrepancy CUD sequences, we can reduce this numerical integration error (the first term) from the standard rate of $O(n^{-1/2})$ to a faster rate of $O(n^{-1+\delta})$ for any $\delta > 0$. However, it is important to note that when using a constant step size $h$ in LMC, the bias term (second term) does not vanish. This bias term includes not only the discretization error of the Langevin diffusion, but also the difference between $f(\theta_k)$ and its truncated version $\bar{f}_\ell(\mathbf{w}_k^{(\ell)})$. Consequently, the bias term in our analysis is larger than the bias term in Vollmer et al. (2016), which employs different techniques and assumptions based on the Poisson equation.

The theorem's assumption of finite Hardy-Krause variation is a common requirement in error bounds for QMC methods, and it can be challenging to verify in practice. Basu and Owen (2016) provide sufficient conditions in order for $f \circ \psi_\ell$ to have finite HK variation, requiring the $\ell$-step transition $\psi_\ell$ to be sufficiently smooth. In the next section, we aim to assess the practical performance of the proposed LQMC algorithm through numerical experiments.

## 5 Numerical experiments

To comprehensively evaluate the performance of the algorithm, we will consider both convex and non-convex potentials, both low-dimensional and high-dimensional state spaces, both accurate and stochastic gradients, both smooth and discontinuous integrands, as well as different learning rate schedules. Additional numerical results can be found in the Appendix B.

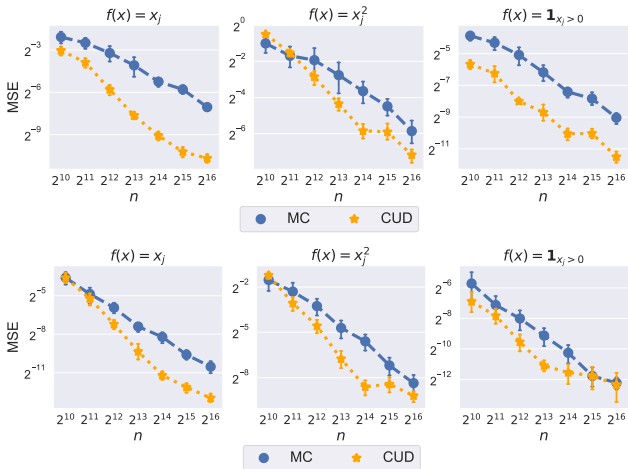

Figure 2: Bayesian logistic regression with accurate gradients (top) and stochastic gradients (bottom).

## 5.1 Bayesian logistic regression

We first consider the Bayesian logistic model

$$y_i \mid x_i \sim \text{Bernoulli}((1 + \exp(-x_i^\mathsf{T}\beta))^{-1}), \quad 1 \leq i \leq N,$$
$$\beta \sim \mathcal{N}(0, I_d).$$

We take $N = 20$, $d = 10$. The features $x_i$ are generated from $\mathcal{N}(0, \Sigma)$ with $\Sigma_{ij} = 2^{-|i-j|}$. The coefficients $\beta$ and the data $y_i$'s are generated from the same model. We consider the test functions $f(x) = x_j, x_j^2, \mathbf{1}_{\{x_j > 0\}}$ for $j = 1, \ldots, d$. The step size $h$ is fixed to 0.001.

We compute the MSE of the estimator based on usual LMC and the proposed LQMC with CUD sequences and report the MSE averaged over all coordinates and 20 random replicates. We do not have a closed form for the expectations $\mathbb{E}[f]$, so the ground truth is estimated using a high-accuracy estimator proposed in He et al. (2023) using scrambled Sobol' sequence with a very large sample size.

In Figure 2 (top panel), we present a log-log plot of the MSE against the number of iterations. Across all three test functions, we observe that LQMC reduces the MSE by a factor ranging from 4 to 8. As the number of iterations increases, the curve corresponding to LQMC reaches a plateau. This behavior can be attributed to the discretization error inherent in the unadjusted LMC, which cannot be further reduced by increasing the number of iterations.

In the bottom panel of Figure 2, we increase the number of observations to $N = 100$ and incorporate stochastic gradient estimation in the Langevin algorithm. Specifically, at each iteration, we estimate the gradient using a random subset of 10 observations. The results demonstrate that LQMC still provides a big improvement when $n$ is smaller than $2^{14}$. However, as $n$ surpasses $2^{14}$, we observe that the LQMC curve flattens again. It is worth noting that the improvement achieved by LQMC in this scenario is less pronounced compared to the previous example, primarily due to the presence of noise in the gradient estimates.

## 5.2 Bayesian linear regression

Now we try a higher-dimensional example with Bayesian linear regression. The model is defined as

$$y_i \sim \mathcal{N}(x_i^\mathsf{T}\beta, \sigma^2 = 4^{-1}), \quad 1 \leq i \leq N,$$
$$\beta \sim \mathcal{N}(0, I).$$

We take $d = 100$ and $N = 20$. We generate $x_i \in \mathbb{R}^d$ similarly as in the logistic regression example. The test functions and step size are also unchanged. The posterior distribution of $\beta$ has the closed form $\mathcal{N}\left((\frac{X^\mathsf{T}X}{\sigma^2} + I)^{-1}\frac{X^\mathsf{T}Y}{\sigma^2}, (\frac{X^\mathsf{T}X}{\sigma^2} + I)^{-1}\right)$. The results are shown in Figure 3. We see that even at 100 dimension, LQMC still brings a substantial improvement over LMC in terms of MSE. In

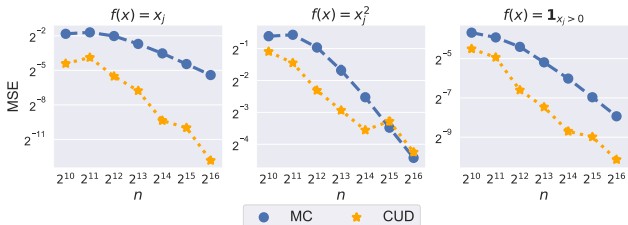

Figure 3: Bayesian linear regression in 100 dimensions.

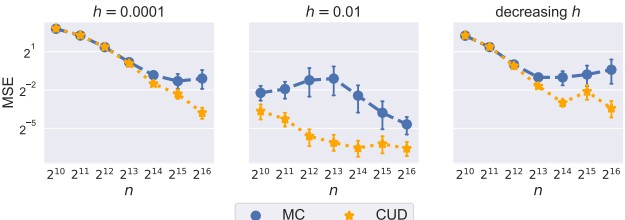

Figure 4: Crossed random effect.

particular, for the integrand $f(x) = x_j$, LQMC achieves a reduction in MSE of approximately 500-fold compared to LMC.

## 5.3   A hierarchical Bayesian model

We consider a hierarchical Bayesian model known as the crossed random effect model

$$Y_{ij} \sim \mathcal{N}(\mu + a_i + b_j, 1), \quad 1 \leq i \leq I, \ 1 \leq j \leq J,$$
$$\mu \sim \mathcal{N}(0,1), \ a_i \overset{iid}{\sim} \mathcal{N}(0, \sigma_a^2), \ b_j \overset{iid}{\sim} \mathcal{N}(0, \sigma_b^2),$$
$$\log(\sigma_a^2), \ \log(\sigma_b^2) \overset{iid}{\sim} \mathcal{N}(0,1).$$

The goal is to sample from the posterior distribution of $(\mu, \mathbf{a}, \mathbf{b}, \log(\sigma_a^2), \log(\sigma_b^2))$, which has dimension $d = I + J + 3$. We take $I = 3$, $J = 5$. We will consider the test functions $f(x) = x_j$ $(1 \leq j \leq d)$. The ground truth of $\mathbb{E}\left[f(x)\right]$ is estimated by Langevin dynamics with Metropolis adjustments (MALA) using a large sample size.

We will compare the performance of the LQMC algorithm using three different step sizes: a constant step size of $10^{-4}$, a constant step size of $10^{-2}$, and decreasing step sizes with $h_k = c_0(c_1 + k)^{-1/3}$. The choice of $c_0$ and $c_1$ ensures that the step size decreases from $10^{-2}$ to $10^{-4}$ throughout the entire algorithm. The use of the exponent $-1/3$ in the decreasing step sizes is recommended in Teh et al. (2016). The results of these comparisons are presented in Figure 4.

In the small step size case (left panel), we observe that the errors of LMC and LQMC are initially comparable for small values of $n$. This is because the algorithm converges slowly, and thus the error is dominated by the bias. However, as $n$ increases, the improvement of LQMC becomes evident. In the large step size case (middle panel), the MSE of LQMC is consistently smaller than that of LMC even for small values of $n$. This is because the algorithm converges faster to the target distribution with a larger step size $h$. Therefore, the improvement of LQMC is more pronounced. Interestingly, in this particular example, using decreasing step sizes yields similar accuracy to using a constant step size of $10^{-4}$. It is worth noting that the MSE of LMC does not decrease at a rate of $n^{-2/3}$ as in Teh et al. (2016). This is because the line in the plot does not represent the accuracy against the iteration $k$ within a single training process. Instead, it reflects the accuracy achieved after completing all $n$ iterations of the algorithm, considering different values of $n$.

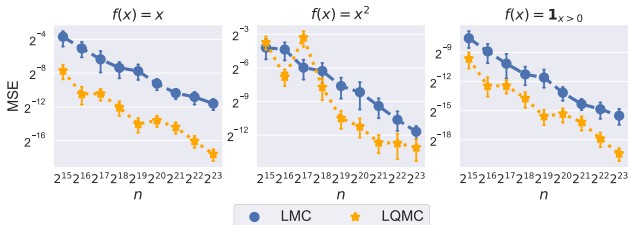

Figure 5: Double well potential.

## 5.4 Nonconvex potential

Finally we investigate a double-well potential function $U(x) = \frac{1}{4}x^2 - \frac{1}{2}\log(1 + x^2)$ from Pagès and Panloup (2018). We know $\mathbb{E}[x] = 0$ and $\mathbb{E}[\mathbf{1}_{\{x>0\}}] = 0.5$. The second moment $\mathbb{E}[x^2]$ is computed by Gaussian quadrature. See the results in Figure 5. Since the potential has two separate local minimums, it takes longer for the Langevin algorithm to explore the space sufficiently and converge to the target distribution. Once converged, the improvement of LQMC over LMC is still significant.

## Acknowledgments and Disclosure of Funding

The author thanks Prof. Art Owen for helpful conversations. This work was partially funded by the NSF grant DMS-2152780 and the Stanford Data Science Scholars program.

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

## A Proofs

### A.1 Proof of Theorem 4.1

We start by decomposing the error $|\frac{1}{n}\sum_{k=1}^{n} f(\theta_k) - \pi(f)|$ into three parts

$$\left|\frac{1}{n}\sum_{k=1}^{n} f(\theta_k) - \pi(f)\right| \leq \left|\frac{1}{n}\sum_{k=\ell+1}^{n} f(\theta_k) - \frac{1}{n}\sum_{k=\ell+1}^{n} \bar{f}_\ell(\mathbf{w}_k^{(\ell)})\right| + \left|\frac{1}{n}\sum_{k=\ell+1}^{n} \bar{f}_\ell(\mathbf{w}_k^{(\ell)}) - \pi(f)\right| + \frac{\ell}{n}2\|f\|_\infty$$

$$= (I) + (II) + \frac{2\ell}{n}\|f\|_\infty.$$

We first upper bound $(I)$.

**Lemma 1** (Upper bound of $(I)$; adapted from Lemma 6.1.4 of Chen (2011)). *If the transition map $\psi$ is a contraction with parameter $\rho$ and if $f$ is 1-Lipschitz, then*

$$|\bar{f}_\ell(\mathbf{w}_k^{(\ell)}) - f(\theta_k)| \leq \left(\max_{0\leq i\leq n}\|\theta_i\| + \mathbb{E}_\pi\left[\|\theta\|\right]\right)\rho^\ell.$$

*Proof of Lemma 1.* Note that

$$|\bar{f}_\ell(\mathbf{w}_k^{(\ell)}) - f(\theta_k)| \leq \int |f(\psi_\ell(\theta, \mathbf{w}_k^{(\ell)})) - f(\psi_\ell(\theta_{k-\ell}, \mathbf{w}_k^{(\ell)}))|\pi(\mathrm{d}\theta)$$

$$\leq \int \|(\psi_\ell(\theta, \mathbf{w}_k^{(\ell)})) - (\psi_\ell(\theta_{k-\ell}, \mathbf{w}_k^{(\ell)}))\|\pi(\mathrm{d}\theta)$$

$$\leq \rho \int \|(\psi_{\ell-1}(\theta, \mathbf{w}_{k-1}^{(\ell-1)})) - (\psi_{\ell-1}(\theta_{k-\ell}, \mathbf{w}_{k-1}^{(\ell-1)}))\|\pi(\mathrm{d}\theta)$$

$$\leq \rho^\ell \int \|\theta - \theta_{k-\ell}\|\pi(\mathrm{d}\theta)$$

$$\leq \rho^\ell(\max_{0\leq i\leq n}\|\theta_i\| + \mathbb{E}_\pi\left[\|\theta\|\right]).$$

$\square$

To bound $(II)$, note that $\frac{1}{n-\ell}\sum_{k=\ell+1}^{n} \bar{f}_\ell(\mathbf{w}_k^{(\ell)})$ is estimating

$$\mathbb{E}\left[\bar{f}_\ell(\mathbf{w}^{(\ell)})\right] = \int \psi_\ell(\theta, \mathbf{w}^{(\ell)})\pi(\mathrm{d}\theta)\mathrm{d}\mathbf{w}^{(\ell)} =: \pi P_\ell(f).$$

Here, $\pi P_\ell$ denote the distribution of the $\ell$-step state $\theta_\ell$ starting from $\theta_0 \sim \pi$. So we have the further decomposition

$$(II) \leq \left|\frac{1}{n-\ell}\sum_{k=\ell+1}^{n} \bar{f}_\ell(\mathbf{w}_k^{(\ell)}) - \pi(f)\right| + \frac{\ell}{n-\ell}\|f\|_\infty$$

$$\leq |\pi(f) - \pi P_\ell(f)| + \left|\frac{1}{n-\ell}\sum_{k=\ell+1}^{n} \bar{f}_\ell(\mathbf{w}_k^{(\ell)}) - \pi P_\ell(f)\right| + \frac{\ell}{n-\ell}\|f\|_\infty$$

$$\leq (II)' + (II)'' + \frac{\ell}{n-\ell}\|f\|_\infty.$$

The first term $(II)'$ is due to the discretization in time. The second term $(II)''$ is the numerical integration error.

To bound $(II)'$, we use the following result.

**Lemma 2** (Upper bound on discretization error $(II)'$). *Under Assumption 1, we have for $f$ 1-Lipschitz,*

$$|\pi(f) - \pi P_\ell(f)| \leq \frac{3\sqrt{2}}{2}\frac{L}{M}h^{1/2}d.$$

*Proof of Lemma 2.* We let $\theta(t)$ be the continuous-time Langevin diffusion with $\theta(0) = \theta_0 \sim \pi$, $W_{t_{k+1}} - W_{t_k} = \sqrt{h}\xi_{k+1}$, where $\xi_{k+1} \overset{iid}{\sim} \mathcal{N}(0, I_d)$, $t_k = kh$. So we have

$$\theta(t_{k+1}) = \theta(t_k) - \int_{t_k}^{t_{k+1}} \nabla U(\theta(s)) \mathrm{d}s + \sqrt{2h}\xi_{k+1}$$

and

$$\theta_{k+1} = \theta_k - h\nabla U(\theta_k) + \sqrt{2h}\xi_{k+1}.$$

Combing the previous two equations gives

$$\theta(t_{k+1}) - \theta_{k+1} = \theta(t_k) - \theta_k - h[\nabla U(\theta(t_k)) - \nabla U(\theta_k)] - \int_{t_k}^{t_{k+1}} \nabla U(\theta(s)) - \nabla U(\theta(t_k)) \mathrm{d}s.$$

Let $\Delta_k = \theta(t_k) - \theta_k$. The last display reads

$$\Delta_{k+1} = \Delta_k - h[\nabla U(\theta_k + \Delta_k) - \nabla U(\theta_k)] - \int_{t_k}^{t_{k+1}} \nabla U(\theta(s)) - \nabla U(\theta(t_k)) \mathrm{d}s.$$

By the contracting property (6) in the main paper,

$$\|\Delta_k - h[\nabla U(\theta_k + \Delta_k) - \nabla U(\theta_k)]\| \leq \rho\|\Delta_k\|.$$

Taking expectation and use $L$-smoothness of $U$, we have

$$\mathbb{E}\left[\|\Delta_{k+1}\|\right] \leq \rho\mathbb{E}\left[\|\Delta_k\|\right] + L\int_{t_k}^{t_{k+1}} \mathbb{E}\left[\|\theta(s) - \theta(t_k)\|\right] \mathrm{d}s.$$

By Lemma 3 of Dalalyan and Karagulyan (2019), $\mathbb{E}\left[\|\nabla U(\theta)\|_2^2\right] \leq Ld$. So we have $\mathbb{E}\left[\|\nabla U(\theta)\|\right] \leq \sqrt{d\mathbb{E}\left[\|\nabla U(\theta)\|_2^2\right]} \leq \sqrt{L}d$. Because $\theta(t)$ is a stationary process,

$$\int_{t_k}^{t_{k+1}} \mathbb{E}\left[\|\theta(s) - \theta(t_k)\|\right] \mathrm{d}s = \int_0^h \mathbb{E}\left[\|\theta(t) - \theta(0)\|\right] \mathrm{d}t$$

$$= \int_0^h \mathbb{E}\left[\| - \int_0^t \nabla U(\theta(s))\mathrm{d}s + \sqrt{2}W_t\|\right] \mathrm{d}t$$

$$\leq \int_0^h \int_0^t \mathbb{E}\left[\|\nabla U(\theta(s))\|\right] \mathrm{d}s\mathrm{d}t + \int_0^h \sqrt{2}\mathbb{E}\left[\|W_t\|\right] \mathrm{d}t$$

$$= \frac{h^2}{2}\sqrt{L}d + \int_0^h \sqrt{2t}\mathbb{E}\left[\|\xi_1\|\right] \mathrm{d}t.$$

Note that

$$\mathbb{E}\left[\|\xi_1\|\right] = \sqrt{2}\frac{\Gamma(d/2 + 1/2)}{\Gamma(d/2)} \leq \sqrt{2}(\frac{d+1}{2})^{1/2} = \sqrt{d+1}.$$

Thus,

$$\int_{t_k}^{t_{k+1}} \mathbb{E}\left[\|\theta(s) - \theta(t_k)\|\right] \mathrm{d}s \leq \frac{1}{2}L^{1/2}h^2 d + \frac{3\sqrt{2}}{2}h^{3/2}d^{1/2}$$

$$\leq \frac{\sqrt{2}}{2}h^{3/2}d + \frac{3\sqrt{2}}{2}h^{3/2}d^{1/2}$$

$$\leq \frac{3\sqrt{2}}{2}h^{3/2}d.$$

Denote $r = \frac{3\sqrt{2}}{2}Lh^{3/2}d$. So

$$\mathbb{E}\left[\|\Delta_{k+1}\|\right] \leq \rho\mathbb{E}\left[\|\Delta_k\|\right] + r \leq \rho^{k+1}\mathbb{E}\left[\|\Delta_0\|\right] + \sum_{i=0}^k \rho^i r$$

$$\leq \frac{r}{1-\rho} = \frac{3\sqrt{2}}{2}\frac{L}{M}h^{1/2}d$$

Therefore, for any $k \geq 1$,

$$|\pi(f) - \pi P_k(f)| = |\mathbb{E}\left[f(\theta(t_k))\right] - \mathbb{E}\left[f(\theta_k)\right]| \leq \mathbb{E}\left[|f(\theta(t_k)) - f(\theta_k)|\right]$$

$$\leq \mathbb{E}\left[\|\Delta_k\|\right] \leq \frac{3\sqrt{2}}{2}\frac{L}{M}h^{1/2}d.$$

$\square$

If we use a noisy gradient $\hat{g}(\theta_k) = \nabla U(\theta_k) + e_k$ where $e_k$ is the noise with mean zero and bounded variance such that $\mathbb{E}(\|e_k\|_2^2) \leq \sigma^2$, then an extra term $2h\sigma$ will appear in Lemma 2. As $\sigma^2$ is usually expected to be proportional to the dimension , this additional term is of the same order as the other term.

**Theorem A.1** (Theorem 9.8 of Niederreiter (1992)). *Let $v_0, v_1, \ldots$ be an LFSR with offset $s$ and period $n = 2^m - 1$ which satisfy $gcd(m, n) = 1$. Then the sequence $\{\mathbf{u}_i\}_{i=0}^{n-1} \subset [0,1]^s$ with $\mathbf{u}_i = (v_i, v_{i+1}, \ldots, v_{i+s-1})$ has, on average, star-discrepancy*

$$O(n^{-1}(\log n)^{d+1}\log\log n)$$

*with an implied constant depending only on $d$ and the average is taken over all primitive polynomials over GF(2) of degree $m$.*

*Proof of Theorem 4.1.* The error on the left-hand-side is bounded by

$$(I) + (II)' + (II)'' + \frac{4\ell}{n}\|f\|_\infty.$$

Lemma 1 shows that $(I) \leq (\max_{0 \leq i < n} \|\theta_i\| + \mathbb{E}_\pi\left[\|\theta\|\right])\rho^\ell \leq (\max_{0 \leq i \leq n}\|\theta_i\| + \mathbb{E}_\pi\left[\|\theta\|\right])h^{1/2}$ since $\ell = \lceil(1/2)\log_\rho h\rceil$. Lemma 2 shows that $(II)' \leq \frac{3\sqrt{2}}{2}\frac{L}{M}dh^{1/2}$. Denote $C_2 = \max_{0 \leq i \leq n}\|\theta_i\| + \mathbb{E}_\pi\left[\|\theta\|\right] + \frac{3\sqrt{2}}{2}\frac{L}{M}d$. So $(I) + (II)' \leq C_2 h^{1/2}$.

By Theorem A.1 and the condition that $\gcd(d\ell, n) = 1$, the star-discrepancy $D^*(\{\bar{w}_k^{(\ell)}\}_{k \geq 1})$ is upper bounded by $O(n^{-1}(\log n)^{d\ell+1}\log\log n)$. Finally, by Koksma-Hlawka inequality, we have $(II)'' \leq \|\bar{f}_\ell\|_{\text{HK}} \cdot D^*(\{\bar{w}_k^{(\ell)}\}_{k \geq 1})$. Thus, $(II)'' + \frac{4\ell}{n}\|f\|_\infty \leq C_1 n^{-1+\delta}$, where $\delta$ hides the poly-logarithmic terms in $\log n$ and $C_1$ depends on $d, \ell, \|\bar{f}_\ell\|_{\text{HK}}$.

Therefore, the upper bound becomes

$$(I) + (II)' + (II)'' + \frac{4\ell}{n}\|f\|_\infty \leq C_1 n^{-1+\delta} + C_2 h^{1/2}.$$

$\square$

# B    Additional numerical results

The primary contribution of this work is to improve LMC as a Monte Carlo sampling algorithm, not as an optimization algorithm. Therefore, our main focus is on providing a better estimation of $\pi(f)$ for some function of interest. Downstream tasks relying on such expectations can also benefit from LQMC. For posterior prediction, it is essential to recognize that the prediction error is not solely determined by the sampling method. Even with infinite perfect samples from the posterior, the prediction error can still arise due to model misspecification, noisy data, biased sampling, etc. So the improvement achieved by LQMC might be less pronounced when assessing the prediction error.

To investigate the performance of LQMC in a posterior prediction setting, we conducted experiments similar to those presented in Dubey et al. (2016) using three UCI datasets. Each dataset was split into a training set (70%), a validation set (10%), and a test set (20%). We performed a tuning process for the constant step size on a grid using the validation set and evaluated the prediction error on the test set. Each iteration computes the stochastic gradient using 32 data points sampled at random. Details of the datasets are in Table 1.

| Datasets | Parkinsons | Bike | Protein |
|---|---|---|---|
| $N$ (number of instances) | 5875 | 17379 | 45730 |
| $p$ (number of features) | 21 | 12 | 9 |

Table 1: Summary of datasets used for Bayesian posterior prediction.

The results are presented in Figure 6. The $x$-axes represent the total number of iterations of Langevin algorithm and the $y$-axes represent the test error. The error bars represent the variation across 10 random replicates. It is evident that LQMC reduced the test error, although the improvement is not substantial. This aligns with our initial expectation, as the proposed method primarily enhances the accuracy of estimating the posterior mean. However, the test error often consists of other sources of error, thus the improvement achieved by the proposed method in reducing the test error might be limited.

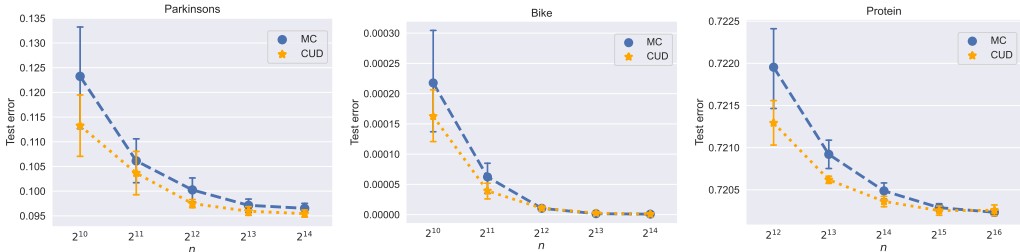

Figure 6: Test error versus number of iterations for the three UCI datasets.

