# OpenReview forum: "Langevin Quasi-Monte Carlo"
_NeurIPS.cc/2023/Conference — NeurIPS 2023 poster_

### Official Review · Reviewer_pgYw · 2023-07-06

**Soundness:** 4 excellent
**Presentation:** 4 excellent
**Contribution:** 3 good
**Rating:** 7
**Confidence:** 4

**Summary:**

This paper analyzes the effect of using quasi-random numbers in place of the usual IID Gaussians for the driving noise of a Langevin algorithm. Assuming that the loss is strong convex and a the quasi-random numbers are completely uniformly distributed, a bound on the Monte Carlo estimation error is derived. This bound is substantially better than what is currently achieved for IID Gaussian driving noise.

**Strengths:**

The paper makes a nice contribution to the theory of sampling with Langevin dynamics, by bringing in the use of quasi-Monte Carlo methods. The proof is quite clean, and it is very clear where the power of the proposed method becomes useful.

The experiments are promising.

**Weaknesses:**

My only real concern is that this paper focuses on a relatively easy case, albeit an important one. It would be interesting to see how easy or difficult it is to extend the results to more complex scenarios, such as with non-convex losses or external random variables.

**Questions:**

* Can the result be extended to common measures between distributions? For example, the result looks almost like a bound on the 1-Wasserstein distance between the distribution of the iterates and the stationary distribution. The main difference is that it assumed that $f$ is both bounded and Lipschitz, as opposed to just Lipschitz.
* Are there general classes of strongly convex functions for which the you can check that the Hardy-Krause variation of $\bar f_{\ell}$ is finite?

**Limitations:**

These are adequately addressed.

---

> ### Author Rebuttal · Authors · 2023-08-08
>
> Thank you for your valuable feedback. We address your questions as follows:
>
> - **Extension to non-convex losses or external random variables** In the "global response", we provided a comprehensive discussion about the strong convexity and smoothness assumptions and discussed to what extent these assumptions might be relaxed. For instance, the strong convexity might be relaxed to functional inequalities, and the Lipschitz gradient might be relaxed to weaker Hölder-continuous conditions. In general non-convex settings, however, the improvement of LQMC might not be as significant, since the advantage of LQMC is more pronounced when the Markov chain mixes well. Nevertheless, in the optimization literature, QMC has demonstrated better exploration capability than traditional Monte Carlo. Hence, it remains an interesting open question to investigate whether LQMC exhibits better exploration than LMC in non-log-concave sampling scenarios. \
> Regarding external variables, our current approach treats the dataset as deterministic, and the posterior distribution is treated no differently from any other target distribution. However, if the dataset is modeled as i.i.d. or a mixing Markov chain, it would be interesting to understand how the sampling of the dataset interacts with the sampling of the Gaussian variables in the LMC algorithm. In the traditional QMC setting, there are studies on combining QMC samples with external samples such as those obtained by acceptance-rejection algorithms or MCMC. How to incorporate external samples into the LMC algorithm is an interesting direction for future work.
> - **Extension to common measures between distribution** To clarify, the question asks whether we can improve the convergence of certain metric such as KL divergence between the law of $x_T$ and the target distribution $\pi$.\
> We do not anticipate the distance between the law of $x_T$ and the target distribution $\pi$ to be smaller by using LQMC as opposed to LMC. In fact, in a well-mixing scenario, the law of $x_T$ obtained by LQMC should be roughly the same as that of LMC.
> However, the key advantage of QMC lies in the collective behavior of the sequence of samples $\{x_1,x_2,\ldots,x_T\}$. Specifically, we expect the empirical distribution $\frac{1}{T}\sum_{k=1}^T\delta_{x_k}$ to be closer to the target $\pi$ in the Kolmogorov-Smirnov distance (also called the star discrepancy). In other words, while each individual sample is not closer to the target, the ensemble of all samples collectively provides a better approximation of the target distribution. As a result, the ergodic average $\frac{1}{T}\sum_{k=1}^T f(X_k)$ provides a more accurate estimate of $\pi(f)$ with LQMC. Furthermore, we believe the ergodic average is of more practical importance since researchers commonly utilize all the samples generated by the Markov chain (excluding potential burn-in or thinning) rather than relying solely on the last individual sample. Thus, the empirical distribution $\frac{1}{T}\sum_{k=1}^T\delta_{x_k}$ is more closely aligned with reality than the distribution of $x_T$.
>
> - **Check bounded variation condition** In response to your question about the bounded Hardy-Krause variation condition, we provided some sufficient conditions and discussions in the "global response".
>
> References:
>
> [1] Hintz, E., Hofert, M., & Lemieux, C. (2022). Quasi-random sampling with black box or acceptance-rejection inputs. In Advances in Modeling and Simulation. Cham: Springer International Publishing.

---

> > ### Comment · Reviewer_pgYw · 2023-08-18
> > **Response**
> >
> > The rebuttal answers my questions adequately. Probably, the most relevant concern of mine was the verification of Hardy-Krause veriation condition, and the global response does a good job of addressing this. I will raise my score by  1.

---

### Official Review · Reviewer_fUNL · 2023-07-06

**Soundness:** 4 excellent
**Presentation:** 4 excellent
**Contribution:** 3 good
**Rating:** 7
**Confidence:** 3

**Summary:**

For suitably smooth functions defined on a bounded support, it is well known that quasi Monte Carlo (QMC) can achieve faster rates of convergence than standard Monte Carlo when the goal is to integrate the function. This paper studies whether techniques from QMC can be beneficial for Langevin Monte Carlo (LMC), a sampling procedure based on the discretization of stochastic differential equations. The paper shows both theoretically and empirically that when the function $f$ is suitably smooth, one can augment the standard Gaussian perturbations in LMC with a specialized sequence such that the error rates converge at a faster rate than standard LMC.

**Strengths:**

The paper is very easy to follow and understand. The authors do a great job of laying out the appropriate context, defining the necessary notions, and laying out their new procedure.

The idea, while relatively simple to implement, seems quite powerful. It’s simplicity seems like a feature rather than bug, and should make it relatively easy to incorporate into open source implementations.

The main result and empirical work seem sound. And while QMC is not new, incorporating it into LMC would be a nice extension of work that has previously been confined to relatively small use cases.

**Weaknesses:**

The main critiques of this paper are associated with its overall applicability. On the theoretical side, the authors make some assumptions to prove their result. How strong are these assumptions? Can they apply generally as the number of samples increases?

In terms of the empirical work, all the examples are done on toy models where the final distributions are known or can be estimated for long chains. Can the authors show LQMC is useful for problems with real data? That would bolster the case for using this as a drop in replacement for vanilla LMC.

**Questions:**

[Q1] The main result requires $\bar{f}_l$ to have bounded variation. Is this possible to show this for any of your examples, i.e. if f is Lipschitz and the target distribution is Gaussian? It would be great to know whether this assumption is weak or strong.

[Q2] Given the main results assumptions on d, m and n, can this result always be applicable for values of n going to infinity?

[Q3] The empirical examples only demonstrate the utility of LQMC on toy models. Can the authors demonstrate its usefulness in a Bayesian context on real data, say for improving the posterior predictions associated with some application?

**Limitations:**

It would be great if the authors mentioned the importance of doing proper inference in ML.

---

> ### Author Rebuttal · Authors · 2023-08-08
>
> Thank you for your valuable feedback. We address your questions as follows:
>
> - **Strong assumptions/bounded variation:**  In the "global response", we have provided a comprehensive discussion on the smoothness, strong convexity, and bounded Hardy-Krause variation assumptions.
> In particular, to address your question regarding the case where $f$ is Lipschitz and $\pi$ is Gaussian, we consider integrands of the form $f\circ \Phi^{-1}:[0,1]^d\to \mathbb R$, where $\Phi^{-1}$ is the component-wise inverse CDF of the standard Gaussian distribution. According to [1], a sufficient condition for QMC to achieve the rate $O(n^{-1+\delta})$ for the integrand $f\circ \Phi^{-1}$ is that, for arbitrarily small $B_i> 0$, there exists $C$ such that
> $$
> |\partial^u f(z)|\leq C\prod_{i=1}^d [1 - \Phi(|z_i|)]^{-B_i}
> $$
> for any $u\subset [d]$.
> This condition implies that as $|z_i|$ tends to infinity, the first-order mixed partial derivatives of $f(z)$ should not grow faster than $1/(1-\Phi(|z_i|))^{B_i}$ for any $B_i>0$. Therefore, when $d>1$, Lipschitz continuity on $f$ is not sufficient nor necessary for this sufficient condition; all first-order mixed partial derivatives $|\partial^u f(z)|$ need to grow relatively slowly.
> - **Can they apply generally as the number of samples increases?** With the increase in the number of samples, the central limit theorem (Bernstein–von Mises theorem) comes into effect, bringing the posterior distribution closer to a normal distribution, where smoothness and strong convexity assumptions hold. However, a challenge arises when utilizing a subsample to estimate the gradient, as the error of the gradient estimator usually scales with the sample size. In such cases, as highlighted by [2], the stochastic gradient's noise dominates, potentially diminishing the applicability of the paper's theorem.
> - **Can the result be applicable for $n$ going to infinity?** In our notation, $n$ is the number of iterations rather than the dataset size. However, we interpret your question as being about the dataset size approaching infinity, and this aspect has been addressed above.
> - **Real example:** We appreciate your interest in real examples and have taken your suggestion into account. We have included new experiments using realistic data and tasks. In particular, we have introduced an example of sparse regression (commonly used for Bayesian variable selection) and several prediction tasks on real datasets. We kindly direct you to the "global response" and the uploaded PDF for detailed descriptions and results of these new experiments.
>
> References
>
> [1] He, Z., Zheng, Z., & Wang, X. (2023). On the error rate of importance sampling with randomized quasi-Monte Carlo. SIAM Journal on Numerical Analysis, 61(2), 515-538.\
> [2] Dalalyan, A. S., & Karagulyan, A. (2019). User-friendly guarantees for the Langevin Monte Carlo with inaccurate gradient. Stochastic Processes and their Applications, 129(12), 5278-5311.

---

> > ### Comment · Reviewer_fUNL · 2023-08-14
> >
> > Thank you for your comments. I have adjusted my score upwards to an "Accept."

---

### Official Review · Reviewer_amBd · 2023-07-06

**Soundness:** 3 good
**Presentation:** 3 good
**Contribution:** 3 good
**Rating:** 7
**Confidence:** 3

**Summary:**

In this paper, the authors proposed the Langevin quasi Monte Carlo algorithm (LQMC), that is, to use a completely uniformly distributed series (CUD quasi random number) in the Langevin Monte Carlo instead of iid pseudo random numbers. The quasi-random number is generated by the LFSR method. For a smooth and strongly convex potential, the estimation error scales as $O(n^{-1 + \delta})$, where $\delta$ encodes the dependence on $\log n$ and $d$. The numerical experiment is performed for various $d$, using gradient or stochastic gradient. In all the cases, LQMC has better performance than the Langevin MC.

**Strengths:**

* Originality: It appears to me that the combination of CUD and Langevin MC is new.
* Quality: This paper contains rigorous derivation of error bounds and numerical experiments covering a wide range of cases. Both derivation and experiment seem convincing for me.
* Clarity: This is a well written paper in general. This paper demonstrates the backbone idea well by using examples and pseudo codes. This paper also provides a good context for the discussion. The setup for the experiments are stated clearly, and the numerical results are interpreted. I also appreciate attaching the code together with the paper.
* Significance: This paper shows that, without making any change to the Langevin MC, changing the random number in the algorithm can lead to vast performance gain. If the performance gain shown in the paper migrates to realistic problems, the precision of a lot of computational work could be improved without a lot of software development.

**Weaknesses:**

* The numerical experiments are performed with synthesized data instead of real ones.
* Unlike classic Monte Carlo, the scaling of error of quasi-Monte Carlo depends on the dimension of the system $d$. As a result, when $n$ is small and $d$ is large, quasi-Monte Carlo may perform worse than classic Monte Carlo in theory. In this paper, the error still depends on $d$ if I am not mistaken. The $d$ dependence is encoded in $\delta$, and not shown explicitly in the main text. Although, in the numerical experiments, it is shown that the LQMC performs very well for $d=100$, I still think it would be nice to make the reader know that, in theory, the error of LQMC depends on $d$.
* The error bound is given in a rather constraint setting.

The weaknesses and questions are addressed by the authors in their rebuttal.

**Questions:**

Overall, I think this is already a very solid paper. However, I think it would be nice if the authors:
* State what this algorithm could do in reality. New experiments using realistic data are welcomed. Discussions on the time scale of the realistic tasks are also interesting (for example, is there any task that was impossible and made possible by LQMC).
* Write the error bound in the format of equation (4), that it, express the $d$ dependence explicitly.

**Limitations:**

My only concern regarding the limitation is that the dependence on $d$ of estimation error is not well studied. After all, $d$ is the criterion of when we should use lattice rules, quasi MC, or MC. However, I understand that the dimension of quasi MC is itself a tricky problem, and, in my humble opinion some ambiguity in this regard could be forgiven.

---

> ### Author Rebuttal · Authors · 2023-08-06
>
>
> Thank you for your valuable feedback. We address your questions as follows:
> - **Numerical experiments are performed with synthesized data:** We understand the importance of real data experiments and have taken your suggestion into account. We have conducted new experiments using realistic data and tasks. Please refer to the "global response" and the uploaded PDF for the detailed description and results of these new experiments.
>
> - **Dimension dependence error for QMC:** We acknowledge that the standard QMC error bound of $O(n^{-1}(\log n)^{d})$ may grow larger than $O(n^{-1/2})$ for large dimensions with insufficient $n$. In practice, however, QMC often performs better than the theoretical bounds suggest. The success of QMC in high-dimensional integration can be attributed to the concept of *effective dimension* [1]. Functions with low effective dimensions can be well approximated by a sum of low dimensional functions, which is favorable for QMC. Moreover, there exist established methods to reduce the effective dimension in the traditional QMC setting, e.g. [2-4]. How to extend these techniques to the Markov chain setting to reduce the effective dimension is an interesting future direction.
> We appreciate your advice, and to highlight the dependence on dimension, we have updated the term $O(n^{-1+\delta})$ to $O(n^{-1}(\log n)^{d})$ in our paper.
>
> - **Error bound is given in a constraint setting:** We acknowledge that the assumptions of strong convexity, smoothness, and bounded variation on the integrand are restrictive. We have discussed these assumptions in the "global response" and mentioned possibilities to relax them. We hope that the theoretical analysis, despite its constraints, provides valuable insights into the performance of our method and its potential for more general scenarios.
>
> References:
>
> [1] Wang, X., \& Fang, K. T. (2003). The effective dimension and quasi-Monte Carlo integration. Journal of Complexity, 19(2), 101-124.\
> [2] Moskowitz, B., & Caflisch, R. E. (1996). Smoothness and dimension reduction in quasi-Monte Carlo methods. Mathematical and Computer Modelling, 23(8-9), 37-54.\
> [3] Imai, J., \& Tan, K. S. (2004). Minimizing effective dimension using linear transformation. In Monte Carlo and Quasi-Monte Carlo Methods 2002 (pp. 275-292).\
> [4] Xiao, Y., & Wang, X. (2019). Enhancing quasi-Monte Carlo simulation by minimizing effective dimension for derivative pricing. Computational Economics, 54, 343-366.

---

> > ### Comment · Reviewer_amBd · 2023-08-16
> >
> > Thank you for the comments and modifications. I would like to keep my rating as "Accept".

---

### Official Review · Reviewer_x8C1 · 2023-07-07

**Soundness:** 4 excellent
**Presentation:** 4 excellent
**Contribution:** 3 good
**Rating:** 6
**Confidence:** 3

**Summary:**

The paper analyses a variant of Langevin Monte Carlo which, instead of using standard Gaussian random variables (using standard random number generation), instead uses correlated random variables following the quasi-MCMC literature. The paper demonstrates that this provably improves the efficiency of certain statistical tests, such as e.g. mean estimation.


**Strengths:**

The usage of non-random Gaussians (using an LSFR sequence instead of standard Mersenne twister) is a novelty for Langevin Monte Carlo. Consequently, the Algorithm is entirely novel, and is provably better in the sense shown in Theorem 4.1. This improvement is a significant technical innovation and is of use to both theoreticians and practitioners.

The experiments are thorough, in that many settings are considered, both in terms of convexity, dimensionality and type of gradient. The improvement for LQMC is clear and significant, which therefore lends credence to the theoretical claims.

The paper is very well written and the proofs are cleanly presented. I could not find any issues with the results.


**Weaknesses:**

The improvement of this method is weak, in the sense that the expectation of certain test functions converges . In contrast, standard Langevin MCMC results will hold in “stronger” measures of convergence, such as KL divergence, total variation, etc. This is expected since quasi-MCMC methods typically can only outperform in these specific circumstances, but is nonetheless a disadvantage of the work.

The analysis seems to be rather trivial and combines the standard inequalities in previous works on quasi-MCMC, with the standard contractivity results of LMC-type algorithms under strong convexity.

Overall, while the analytic simplicity of this paper makes the result seem obvious in hindsight, I would still argue that this result is novel and impactful enough to merit publication.


**Questions:**

Are the assumptions of smoothness, strong convexity and bounded variation (of f) necessary? I would recommend a deeper exploration of these conditions in order to enrich the paper.

Although stochastic gradient LMC appears in the experiments, it is not analysed. Could similar results be established in this setting?

Could this approach be combined with e.g. Metropolis/HMC algorithms, or other MCMC methods based on similar walks? It seems there may be some significant theoretical barriers to this end.

Typos:
Some citations are not properly formatted, e.g. in L. 22, L. 126, L. 151 (using citet instead of citep or vice versa).


**Limitations:**

None beyond those raised in my earlier comments.

---

> ### Author Rebuttal · Authors · 2023-08-07
>
>
> Thank you for your valuable feedback.
>
> We appreciate your recognition regarding the differences in our analysis compared to the standard literature on Langevin Monte Carlo (LMC). Traditional results in LMC primarily focus on studying convergence through metrics like the KL divergence between the law of $x_T$ and the target distribution $\pi$, thus investigating the distributional properties of the sample drawn at the $T$-th iteration in the LMC algorithm. In contrast, our analysis centers on the convergence of the ergodic average $\frac{1}{T}\sum_{k=1}^T f(x_k)$ to the target $\pi(f)$. This perspective directly examines the behavior of the average function evaluations over the LMC samples, providing a more direct evaluation of the quality of LMC in estimating expectations. While our approach differs from the conventional ones, we believe that our result is not inherently weaker for the following reasons:
> 1. Firstly, the convergence rate of the law of $x_T$ does not necessarily imply the convergence rate of the ergodic average $\frac{1}{T}\sum_{k=1}^T f(x_k)$.
> 2. Furthermore, our result, while stated as the convergence of expectation, also implies the convergence of the empirical distribution $\frac{1}{T}\sum_{k=1}^T\delta_{x_k}$ to the target distribution in the sense of the Kolmogorov-Smirnov distance (i.e., star discrepancy). In fact, this convergence of star discrepancy is the reason why we can achieve smaller errors for integrands of bounded variation in the sense of Hardy and Krause.
> 3. Lastly, in practical applications, researchers often utilize all the LMC samples obtained (excluding possible burn-in and thinning) rather than focusing solely on the last single sample. Consequently, the quality of the empirical distribution $\frac{1}{T}\sum_{k=1}^T\delta_{x_k}$ is of greater practical relevance than the law of $x_T$.
>
> While we acknowledge that our analysis introduces a nonstandard technical condition of bounded variation and establishes a different sense of convergence, we do not perceive this as a weakness. On the contrary, we believe this nonstandard approach enriches the standard literature and offers a new perspective to the understanding of LMC.
>
> Below is a point-by-point response to your questions:
> - **Are the assumptions necessary?** Please refer to the "global response" for a discussion on the assumptions of smoothness, strong convexity, and bounded variation. In particular, we discussed the possibilities to relax the assumptions of smoothness and strong convexity, and provided some sufficient conditions to check bounded HK variation.
> - **Analysis for stochastic gradients:** We appreciate your suggestion. If we use a noisy gradient $\hat g(\theta_k)=\nabla U(\theta_k)+e_k$ where $e_k$ is the noise with mean zero and bounded variance such that $\mathbb{E}(||e_k||_2^2)\leq\sigma^2$, then an extra term $2h\sigma$ will appear in Lemma 2 in the proof. As $\sigma^2$ is usually expected to be proportional to the dimension $d$, this additional term is of the same order as the other term. If the stochastic gradient $\hat g$ is estimated by a subsample, then $\sigma^2$ might grow with the sample size of the dataset, making this extra term dominate, as pointed out in [1]. We have included this analysis in the revision.
> - **Combining with Metropolis/HMC algorithms:** Algorithmically, incorporating a rejection step is straightforward by generating an additional uniform random variable. However, one crucial consideration that prevents us from adopting a rejection step in this work is that the rejection step introduces discontinuities, which is not QMC-friendly. In contrast, the unadjusted Langevin algorithm (ULA) is continuous in the underlying uniform random variables, making it more favorable for QMC. Beyond the issue of discontinuities, the rejection step may lead to slow mixing when the rejection rate is high. Thus, while we acknowledge the straightforward implementation of a rejection step, we agree with you that there may be theoretical barriers to this end.
> - Thank you for bringing the typos to our attention. We have addressed them in the revision.
>
> [1] Dalalyan, A. S., & Karagulyan, A. (2019). User-friendly guarantees for the Langevin Monte Carlo with inaccurate gradient. Stochastic Processes and their Applications, 129(12), 5278-5311.

---

> > ### Comment · Reviewer_x8C1 · 2023-08-20
> > **Response**
> >
> > I thank the authors for their detailed responses. The authors presented a fair justification for the chosen measure of convergence, which was one of my primary criticisms. I am choosing to maintain my current score, which was already relatively high.

---

### Author Rebuttal · Authors · 2023-08-06

# Global response
We thank the reviewers for their insightful and constructive feedback. The positive evaluations of our work as "novel, impactful, significant, promising, and important" are encouraging, and we appreciate the recognition of the clarity and cleanliness of the paper. In this global response, we address the common aspects raised by multiple reviewers regarding the strong assumptions in the theoretical analysis and the absence of real data examples. Specific response to individual reviewers are provided separately.

## Strong convexity and smoothness
- We acknowledge that these are restrictive assumptions. However, starting with these assumptions is a necessary and natural first step to study the proposed LQMC algorithm. Evaluating its performance in the simplest setting allows us to gauge its potential for more complicated scenarios.
- The simplicity of the analysis yields valuable insights into how the driving uniform variables impact LMC. The error decomposition provides insights regarding which aspect of the algorithm might benefit from QMC. Despite its simplicity, the theoretical analysis guides the design of uniform numbers and sets expectations for the algorithm's performance, which might not be as clear in a more general setting.
- Next, we recognize the opportunity to relax these assumptions. Techniques similar to [6] can be adopted in non-strongly convex settings by introducing a quadratic penalty. A broader class of measures satisfying log-Sobolev or Poincaré inequalities also presents convergence guarantees [7]. Relaxing the Lipschitz gradient to a weaker Hölder-continuous gradient might be possible [5]. Growth conditions like dissipativity might also ensure convergence [9]. Quantifying the improvement of LQMC in these more general yet tractable settings is an interesting future direction, albeit beyond the scope of this work.
- While the improvement of LQMC might not be significant in general non-convex settings, where only first-order stationarity guarantees can be expected, the question of whether LQMC enhances exploration in non-convex settings remains open. Encouragingly, studies in optimization demonstrate QMC's superior exploration properties, such as in reinforcement learning, where QMC outperforms iid Gaussian for parameter exploration [10], and in variational inference, where QMC converges faster compared to traditional Monte Carlo [11]. These findings highlight QMC's potential benefit for exploration.

## Bounded Hardy-Krause variation
The condition of bounded HK variation is indeed a standard condition in QMC theory, ensuring the error rate of $O(n^{-1+\delta})$ for any $\delta>0$, with log terms hidden. While verifying this condition is challenging in general, there exist sufficient conditions [1,2]:
$$V_{HK}(f)\leq\sum_{u\subset1:d,u\neq\emptyset}\int_{[0,1]^{|u|}}|\partial^uf(x_{u},1_{-u})|dx_u.$$
In our analysis, we require $\bar f_{\ell}=f\circ g$ to have bounded HK, where $g=\psi_{\ell}:[0,1]^d\to G$ is the $\ell$-step transition. Consider two cases:
- **$G$ is compact:** If all the mixed partial derivatives of $g$ are square integrable, then $f\circ g$ has bounded HK for any $f\in C^d(G)$ [1]. Hence, if LMC samples are constrained within a compact set and the transition is sufficiently smooth, then $\bar f_\ell$ has bounded HK for any $f\in C^d$.
- **Otherwise:** If the function grows not too rapidly on the boundary of the unit cube, QMC can still achieve the error rate of $O(n^{-1+\delta})$. Some boundary growth conditions are introduced in [8] to ensure this rate.

Verification of these sufficient conditions is easier if such mixed partial derivatives are bounded. In other cases, one needs to examine the function's growth at singularities. It is essential to note that the bounded variation condition is a technical requirement for QMC theory. Empirically, QMC can still outperform Monte Carlo without this condition, as its performance largely depends on factors like smoothness and effective dimension.

## Real data experiments
Real data experiments were conducted in response to reviewers' suggestions. We first emphasize that the primary contribution of this work is to *improve LMC as a Monte Carlo sampling algorithm, not as an optimization algorithm*. Therefore, our main focus is on providing a better estimation of $\pi(f)$ for some function of interest. Downstream tasks relying on such expectations can also benefit from LQMC. For posterior prediction, it is essential to recognize that the prediction error is not solely determined by the sampling method. Even with infinite perfect samples from the posterior, the prediction error can still arise due to model misspecification, noisy data, biased sampling, etc. So the improvement achieved by LQMC might be less pronounced when assessing the prediction error.

The results are presented in the uploaded PDF. The additional experiments provide a more comprehensive evaluation of LQMC's effectiveness in realistic tasks.

[1] Basu et al. (2016). Transformations and HK variation. SINUM.\
[2] Owen. (2005). Multidimensional variation for QMC. Contemporary Multivariate Analysis And Design Of Experiments.\
[3] Dalalyan et al. (2012). Sparse regression learning by aggregation and LMC. Journal of Computer and System Sciences.\
[4] Dubey et al. (2016). Variance reduction in SGLD. NeurIPS.\
[5] Chatterji et al. (2020). LMC without smoothness. AISTATS.\
[6] Dalalyan et al. (2022). Bounding the error of discretized Langevin algorithms for non-strongly log-concave targets. JMLR.\
[7] Chewi et al. (2022). Analysis of LMC from Poincare to Log-Sobolev. COLT.\
[8] Owen. (2006). Halton sequences avoid the origin. SIAM Review.\
[9] Erdogdu et al. (2021). On the convergence of LMC: The interplay between tail growth and smoothness. COLT.\
[10] Choromanski et al. (2018). Structured evolution with compact architectures for scalable policy optimization. ICML.\
[11] Buchholz et al. (2018). QMC variational inference. ICML.

---

### Decision · Program_Chairs · 2023-09-21

**Decision:**

Accept (poster)

**Comment:**

This paper considers sampling with a variation of Langevin Monte Carlo algorithm. Authors suggest using correlated random variables as in the quasi-Monte Carlo framework. Authors show that this improves the performance of vanilla LMC in which standard Gaussian random variables are used.

The paper is reviewed by 4 expert reviewers. The Rating/Confidence  scores are 6/3, 7/3, 7/3, 7/4 and reviewers all agree that the paper is well-written with significant contributions. I recommend accepting this paper to the conference program.

I recommend authors to incorporate the relevant part of the discussion period in their camera ready version. In particular, a substantial portion of the recent literature is not mentioned in the paper.